# Analysis of the Network Efficiency of Chinese Ports in Global Shipping under the Impacts of Typhoons

**Tianni Wang [1], Haochen Feng [1], Mark Ching-Pong Poo [2],* and Yui-Yip Lau [3]**

[1] College of Transport & Communications, Shanghai Maritime University, Shanghai 201306, China; wangtn@shmtu.edu.cn (T.W.); 202230610151@stu.shmtu.edu.cn (H.F.)
[2] Liverpool Hope Business School, Liverpool Hope University, Liverpool L16 9JD, UK
[3] Division of Business and Hospitality Management, School of Professional Education and Executive Development, The Hong Kong Polytechnic University, Hong Kong, China; yuiyip.lau@cpce-polyu.edu.hk
* Correspondence: pooc@hope.ac.uk

**Abstract:** With the increasing volume of international trade and maritime demand, the requirements for the stability and reliability of the global shipping system are also increasing. The research on the network efficiency of Chinese ports for global shipping can not only examine the importance of Chinese ports in the shipping network but also find out the aspects that need to be improved in the construction of the port's climate adaptability in the resilience assessment to strengthen port construction and further improve the efficiency of the network. The current study builds a shipping network based on RCEP and systematically examines the key ports in China within the networks. The research paper aims to improve the resilience of the ports and the whole shipping network in response to typhoon disasters. As such, this paper focuses on shipping research based on complex networks and network multi-centricity analysis, followed by a ranking of ports. Firstly, this paper uses UCINET 6 software to build a global shipping network. Such a network evaluates the centrality of ports, calculates the degree of centrality, proximity to centrality, and centrality, and scores them according to the ranking. Then, it selects the top 20 ports in China according to the ranking and researches network efficiency for the listed ports considering the typhoon risks. The analysis of network robustness, average shortest path length, and network efficiency are carried out for the shipping network and China's essential port nodes in the network. According to the experimental results, no matter the robustness, average shortest path length, or network efficiency, when the important ports of China in the shipping network are affected, they will cause different degrees of impact, and the performance loss caused by multiple ports is higher than that of a single port. They emphasise the significant impact of typhoons on multiple ports and remind people to minimise losses as much as possible based on experimental results, ensuring the stable operation of ports and improving resilience in typhoon prevention under the changing climate. Additionally, they provide a solid foundation to further sustain global shipping network resilience.

**Keywords:** port resilience; network efficiency; typhoon assessment

## 1. Introduction

According to data from China Customs in 2019, 80% of China's import and export trade is carried out through sea transportation, and the stability of sea transportation determines whether the trade can be carried out in an orderly manner. However, the ports in the east and southeast of China are highly susceptible to typhoons from July to September, with around seven typhoons affecting China's ports yearly [1,2]. According to the data of the China Marine Disaster Bulletin 2022, the direct economic losses caused by typhoons and storm surges along the coast of China in the past decade reached RMB 6.8 billion. In addition to the economic losses, China's containerised port throughput accounts for about 30% of the global port throughput [3] and plays a pivotal role in the

global containerised trade system. Therefore, when a typhoon affects the ports, it could directly lead to port closure, affect port operations and cargo transportation efficiency, and ultimately deteriorate the supply chain in which the port is located, reducing the efficiency of international trade.

Due to the nature of the high risk of typhoons and other meteorological disasters, as well as the significant losses caused by them, Chinese national and local governments have diverse typhoon prevention and flood control contingency plans to enhance the emergency response capacity of the port and to strengthen the daily publicity and disaster preparedness drills to improve the awareness of the port staff for the prevention of disasters.

Based on a literature review, the authors found that to reduce the significant impact of typhoons on ports, scholars have researched how ports can improve the response to typhoon disasters from different perspectives, which mainly concern the following three aspects: port vulnerability, port adaptation measures, and port resilience. The resilience of the shipping network reflects the change in the whole process of the shipping network after it has been affected by external influences by which the changes in the performance of the shipping network can be reflected. Lau et al. [4] criticised past research studies (e.g., [5]) that only addressed pure theoretical or conceptual matters, proposing that their relevant findings may not generate an unbiased vortex with physical reliability. Some researchers concentrated on adopting complex numerical weather models to predict weather impacts [6–8]. Moreover, most research studies on tropical cyclones incline towards a single case study, like Taiwan [9], New York City [10], and Macau [11]. Nevertheless, few studies have considered how typhoon disasters affect shipping network resilience.

The Regional Comprehensive Economic Partnership (RCEP), entering into force in China in January 2022, is the largest and most important free trade agreement negotiated in the Asia–Pacific region. It is one of the most dynamic free trade areas in the world and the largest and most important free trade agreement negotiated in the Asia–Pacific region, covering more than 3.5 billion people, accounting for 47.4% of the world's population, 32.2% of the world's gross domestic product, and 29.1% of the world's total foreign trade. The agreement includes China, Japan, South Korea, Australia, New Zealand, and the ten ASEAN countries as its fifteen members, which significantly promotes the confidence and determination of all countries in adhering to multilateralism and free trade and promoting regional economic integration. The implementation of the RCEP provides a crucial institutional guarantee for member countries to achieve long-term prosperity, stability, and high quality of life for human beings [12].

This paper constructs a shipping network based on the RCEP and systematically analyses the crucial ports in China within the networks. By comprehensively investigating the economic losses, repair time of facilities after a disaster, and corresponding improvement measures for the weak links therein, it aims to enhance the resilience of the ports and the whole shipping network.

## 2. Literature Review

This literature review offers a detailed examination of the use of complex network theory in shipping network studies, showcasing an array of investigations focused on the structural attributes, resilience, and susceptibility of maritime transport systems. Utilising graph theory and network models, researchers have explored container transportation patterns and the vulnerability of cargo flow across global and regional scales. Investigations extend into analysing synergistic relationships within container transportation networks and evaluating network nodes' importance through novel methods. Significant attention has also been given to the shipping networks' robustness, destructiveness, and vulnerability, providing insights into the networks' resilience against deliberate disruptions and natural calamities. Furthermore, the review delves into typhoon risk assessment and mitigation strategies, highlighting the crucial need for understanding and lessening typhoon impacts on port operations. Despite progress in assessing the immediate consequences of such

natural disasters, a research gap is evident concerning the recovery processes of shipping networks post-typhoon.

### 2.1. Shipping Research Based on Complex Networks

At present, many scholars have utilised complex networks as the theoretical basis for the study of shipping networks. Laxe, for instance, based on the theoretical basis of graph theory, studied the inter-port and inter-region container transportation patterns and changes [13]. Chen employed multiple centrality analyses in complex networks to conduct the research; the former focused on the change in port ranking after the network was affected by the epidemic. In contrast, the latter focused on the characteristics of Southeast Asian shipping networks through centrality analysis [14]. In the meantime, Mou et al. [15] and He et al. [16] conducted experimental tests for the destructive resistance of the shipping network. The former used a complex network analysis method with directed weights to study the characteristics and destructive resistance of the shipping network in terms of the topology and the performance of the shipping network. The latter established a load redistribution strategy based on the residual capacity of the neighbouring points of the faulty node, constructed a dynamic load network model containing load fluctuations, and used the interval optimisation method to study the influence of each parameter in the model on the network's destructive resistance.

Wu et al. [17] established a relationship model between significant ports along the "Maritime Silk Road" and studied its statistical characteristics, such as the average path length, agglomeration coefficient, degree, and distribution. Based on the above analysis, a set of comprehensive port centrality assessment methods was established by combining the Byrd counting method with degree centrality, intermediate centrality, and proximity centrality. In addition, Wan et al. [18], through constructing the Maritime Silk Road shipping network, conducted an in-depth study on the resilience assessment of the maritime transportation system and established a quantitative index of resilience according to the resilience characteristic curve based on the theory of the "resilience triangle". The resilience of the shipping network in disasters was assessed, and countermeasures and recommendations were proposed to cope with the impacts of disasters through a case study.

Most research on maritime transportation networks focuses on the characteristics and pattern changes of the shipping network. Others are concerned about the shipping network's destructiveness, robustness, and vulnerability, including the performance change in the shipping network affected by external factors and countermeasures to improve the destructive absorption capacity of the shipping network accordingly. Nevertheless, the previous research mainly investigates the absorption stage of the shipping network but usually ignores the recovery phase influenced by external factors.

### 2.2. Shipping Research Based on Typhoon Risks

In studies related to typhoon risk assessment, scholars utilised historical data and software to simulate the impacts caused by wind fields. Jian et al. [19] developed a cyclone risk model for container ports using a holistic approach to assess the physical vulnerability of port infrastructure and cargo to cyclone events. Lam et al. [20] developed a scientific cyclone risk mapping methodology for critical coastal infrastructure in the case of East Asia seaports. The mapping results showed that cyclone events were quite high in the coastal waters extending from Vietnam to Japan. In Southeast Asia, cyclone risk was generally low in terms of wind intensity and frequency. In Northeast Asia, the ports of Shanghai, Kaohsiung, and Keelung were more vulnerable to cyclone risk. Combining the MIKE21 spectral model with the Holland wind field model, Hou et al. [21] simulated the wave characteristics of the fishing port of Canmen under Typhoon Lekima's influence and evaluated the port's berthing capacity.

Wang et al. [22] employed the cross-corrected multi-platform ("CCMP") wind field data as the database to reconstruct the wind field of the Yellow Sea and Bohai Sea for 30 years from 1991 to 2020. The experimental results showed potential marine disaster risks

existed in Xinjiekou, Daihekou, Yanghekou, and Shanhaiguan fishing harbours. Similarly, Hong and Ji [23] utilised the typhoon meteorological observations from 1997 to 2015 and the statistical data of asset distribution in the Nansha harbour area of Guangzhou City and adopted the International Strategy for Disaster Reduction (ISDR) assessment model to quantitatively evaluate the typhoon risk in Nansha harbour in different recurrence periods. In the case of Shanghai Yangshan Port, Liu et al. [24] propose a fine-scale coastal storm surge vulnerability and risk assessment model, considering exposure of disaster-bearing bodies, sensitivity, and adaptability indicators. Application in Laizhou Bay, China, highlights areas at higher risk, emphasising the model's flexibility and multi-scale assessment capabilities.

Zhang et al. [25] used geospatial techniques and fuzzy evaluation principles to quantify typhoon storm surge (TSS) risk in 14 coastal cities of Guangdong Province. Results identified significant risk zones, with targeted disaster prevention strategies proposed, emphasising the need to improve mitigation capacity in high-risk coastal regions and implement exposure reduction measures in vulnerable areas. On the other hand, Ding and Dong [26], based on data from 33 coastal counties in Fujian Province from 2011 to 2020, scientifically assessed the spatial and temporal differentiation characteristics of the resilience of coastal cities. The urban safety resilience triangle model was utilised to carry out comprehensive risk zoning for storm surge disasters in Fujian Province, synthesising the evaluation of storm surge disaster risk, vulnerability, and urban toughness. Ding and Wei [27] proposed a nested model based on the three-dimensional finite-volume coastal ocean model that successfully simulates tide and storm surge heights in the Bohai Sea, China, closely matching observed data from coastal tide gauges. Sensitivity experiments indicate that coastline changes due to land reclamation can significantly influence water levels during storm surges, potentially increasing maximum water levels by 0.1–0.2 m at significant ports, posing a risk of severe damage. In addition to risk assessment, many scholars have focused on reducing typhoons' impact on ports to minimise losses, such as through better ship scheduling and wind protection measures for harbour machinery. Zhang et al. [28] indicated that the Taihu Basin, a vital economic core in the Yangtze River Delta, faces significant flood risks due to rapid urbanisation and environmental degradation. This paper analyses flood risk characteristics, identifies existing challenges in the flood control system, and proposes structural and non-structural measures to address these issues.

In response to the demand for anchorage, Guangyi et al. [29] optimised the construction area of prevention anchorage by considering the construction and maintenance cost and the cost of ship evacuation and typhoon avoidance routes based on joint scheduling. It, therefore, ensured the safety of the whole waterway transportation system and reduced the construction and maintenance costs. By analysing the necessity of disaster prevention and mitigation in harbour and navigation engineering, Zhang et al. [30] discussed the specific measures of disaster prevention and mitigation in harbour and navigation engineering from diverse aspects. Furthermore, Lin and Lu [31] developed windproof safety systems for large-scale port equipment in response to the growing demand for handling offshore wind power components, which pose unique challenges due to their scale and vulnerability. Their study establishes a risk analysis framework providing valuable insights and critical risk factors for safety management in offshore wind handling operations, particularly relevant for emerging offshore wind markets facing similar challenges. On the other hand, the uncertainty of freight demand can also be influenced by climate change, such as the change in demand and supply for the products, indicating a change in freight demand [32]. Therefore, the integration of sustainability practices within freight transportation not only addresses the immediate environmental and societal challenges posed by the industry but also fortifies the supply chain's resilience against the unpredictable impacts of climate change on demand and supply dynamics [33].

In the above studies, some looked at the typhoon disaster and studied its wind field characteristics, while others assessed typhoons in ports to identify the high-risk areas according to the historical typhoon data. Presently, the research on wind protection

measures for ports mainly focuses on the levels of ship wind avoidance and mechanical wind protection and on what kind of equipment or parts are to be used to improve the wind protection ability of the basic hardware facilities of ports.

Unfortunately, there is a lack of research on how the shipping network changes after the port has suffered from the impact of typhoons. It is more conducive to discovering the various deficiencies of the network in the complete process of being affected and improving and perfecting it more comprehensively. In addition, as it is difficult to quantify the recovery effect of various measures, there are few studies on improving network recovery efficiency to enhance network resilience. This paper constructs a shipping network based on the complex network theory and examines the varieties of network efficiency after typhoon events occur.

## 3. The RCEP Shipping Network

This section may be divided by shipping research based on complex networks and network multi-centrity analysis, followed by a ranking of ports.

### 3.1. The Network Construction

According to complex network theory, the network consists of point sets and connecting edges. The ports are abstracted as nodes in the network, the collection of multiple ports is the network point set, and the routes between ports are regarded as network connecting edges.

There are two main ways of constructing complex networks: the direct connectivity graph and the complete connectivity graph. The direct connectivity graph is that a port is only connected to neighbouring ports; there is no intermediate port between ports, and only neighbouring ports can be reached, while the view in the complete connectivity graph is that ports can be reached regardless of their proximity, that is, it is possible to start from the port of origin, pass through one or more intermediate ports, and finally arrive at the port of destination. It can be found that the composition of the complete connected diagram coincides with the actual ship navigation situation, that is, the container liner sails between various ports by the established route. In addition, the complete connected graph does not consider the transportation distance between ports and pays more attention to the importance of nodes in the network and the hub role of ports, which is also more conducive to observing the performance change in the whole network after the failure of important nodes. Thus, this section utilises the composition of a complete connected graph to construct a shipping network.

The signing of the RCEP Agreement involves fifteen countries, namely, China, Japan, Korea, Australia, New Zealand, Indonesia, Malaysia, the Philippines, Singapore, Thailand, Vietnam, Laos, Myanmar, Cambodia, and Brunei. Therefore, this paper establishes a shipping network through the routes of these 15 countries. To make the established shipping network closer to reality, the statistics of Asian routes and Asia–Australia routes of the world's top 100 shipping lines are based on the official website of Alphaliner, a French shipping analysis company, in which the top 10 shipping lines have a total capacity of 23,140,000 TEUs, which occupies 84.4% of the total capacity in the world. It can be seen that the shipping network constructed by utilising the route data of the top ten shipping companies has a certain degree of objectivity and is closer to reality.

In this paper, an 82 × 82 0-1 adjacency matrix was constructed using UCINET software (https://sites.google.com/site/ucinetsoftware/home, accessed on 1 March 2024). Thus, the RCEP shipping network was established based on the composition of a fully connected graph, as shown in Figure 1. The network graph has a total of 82 port nodes and 866 connected edges. It can be seen in the figure that the constructed shipping network is complex, with many connecting edges and better connectivity, and the individual nodes have more connecting edges, which also indicates that the ports represented by the nodes are involved in more routes, with high accessibility and a stronger role as hubs.

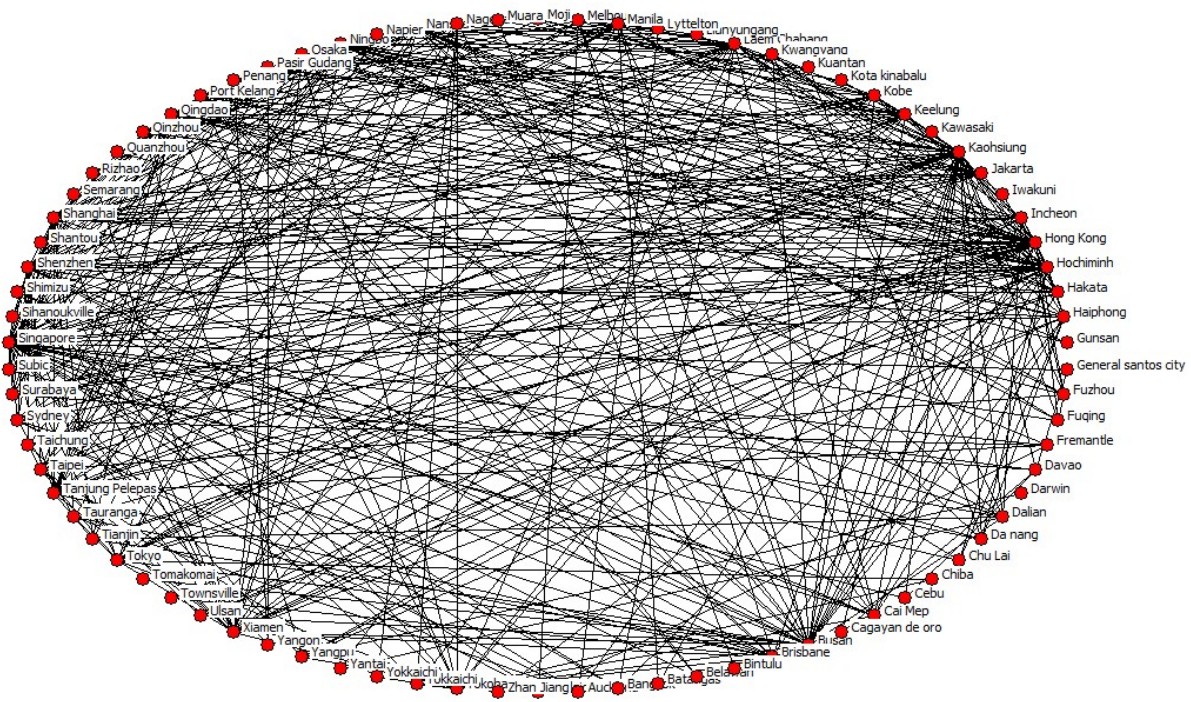

**Figure 1.** RCEP shipping network diagram.

*3.2. Network Multi-Centricity Analysis*

The upcoming analyses can be split into a few parts: degree centrality, proximity centrality, intermediate centrality analysis, and core-edge analysis.

### 3.2.1. Degree Centrality Analysis

The degree of centrality value is the most intuitive indicator for describing the centrality of nodes in the network. The higher the number of connections between a node and neighbouring ports, the higher the degree of centrality of the port. This means that the communication ability between ports is stronger, the accessibility is stronger, and the importance of the port is greater. The formula for calculating the degree centrality value is shown in Equation (1), where n refers to all the port nodes in the network and $\delta_{ij}$ is the number of connections between port *i* and port *j*.

$$D_i = \sum_{j}^{n} \delta_{ij} \tag{1}$$

As shown in Table 1, the Chinese ports ranked in the top 10 in the RCEP shipping network are the ports of Kaohsiung, Hong Kong, Shenzhen, Shanghai, Ningbo Zhoushan, and Qingdao, and the average number of connections between these ports and other ports is 31.17, which is much higher than that of the overall network (10.59). These data indicate that they have a stronger ability to connect with other ports than other ports in the RCEP shipping network and have a stronger hub capacity than other ports. On the other hand, ports with a low degree of centrality are predominantly destination ports and do not have a strong hub function. In addition, the remaining four ports among the top ten ports are located in Southeast Asia and South Korea. It can be seen that China's ports occupy the most important position in the RCEP shipping network and play a pivotal role in the operation of the entire shipping network.

**Table 1.** Degree centrality rank.

| Port | Degree | Regional Rank | Network Rank |
|------|--------|---------------|--------------|
| Kaohsiung | 37 | 1 | 2 |
| Hong Kong | 35 | 2 | 3 |
| Shenzhen | 31 | 3 | 4 |
| Shanghai | 30 | 4 | 5 |
| Ningbo | 28 | 5 | 8 |
| Qingdao | 26 | 6 | 9 |

3.2.2. Proximity Centrality Analysis

Proximity centrality is the reciprocal of the sum of the shortest distances from a node to a target node, reflecting the proximity, i.e., global reachability, between a node of the network and other nodes of the network. Degree centrality only considers the number of connections between a node and its neighbouring nodes. In contrast, proximity centrality is considered a global attribute, which can better evaluate the degree of superiority of the node in the network spatial location. If the shortest distance from the node to all other nodes in the network is very short, then its proximity centrality is high, which means that the higher the accessibility of the node, the higher the possibility of being a hub port and the higher the importance of the node in the shipping network. The proximity centrality formula is shown in Equation (2), where $n - 1$ represents the total number of nodes of other ports in the target port and $\delta_{ij}$ represents the number of connections between port $i$ and port $j$.

$$D_{C_i} = \frac{n - 1}{\sum_j^n \delta_{ij}} \tag{2}$$

In Table 2, the calculation results show that the ports with higher proximity centrality are Kaohsiung Port, Hong Kong Port, Shanghai Port, Shenzhen Port, Qingdao Port, Ningbo Zhoushan Port, and Xiamen Port. When the proximity centrality of a node is higher, its connection with other ports is also higher. This means that these ports are also more critical in the network. Compared with the degree centrality ranking in the previous section, it can be found that the three ports of Kaohsiung, Hong Kong, and Shanghai ports occupy a critical position in both the local network and the overall network and play a prominent role as hubs in the network. Overall, the difference in proximity centrality between ports is relatively small, and most of the ports have high values. Only 25% of the ports have proximity centrality values below 40, which indicates that the accessibility of the ports in the network is better. The fact that Chinese ports account for 70% of the top ten ports also indicates that Chinese ports have better spatial advantages in the network, with more ports having a hub role.

**Table 2.** Closeness centrality rank.

| Port | Closeness Centrality | Regional Rank | Network Rank |
|------|----------------------|---------------|--------------|
| Kaohsiung | 63.281 | 1 | 2 |
| Hong Kong | 61.832 | 2 | 3 |
| Shanghai | 59.559 | 3 | 4 |
| Shenzhen | 58.273 | 4 | 6 |
| Qingdao | 57.042 | 5 | 7 |
| Ningbo | 56.643 | 6 | 9 |
| Xiamen | 55.862 | 7 | 10 |

3.2.3. Intermediate Centrality Analysis

Intermediate centrality refers to the ratio of the shortest paths passing through a certain point in the network and connecting these two points to the total number of shortest paths between these two points, reflecting the communication efficiency between port nodes and other port nodes, i.e., the role of the "middleman". The formula for calculating the

centrality of the intermediary is shown in Equation (3), where $\delta(s,t|i)$ is the number of shortest paths of ports $s$ and $t$ through port $i$ and $\delta(s,t)$ is the sum of the shortest paths of ports $s$ and $t$.

$$B_{C_i} = \sum_{s,t \in v \ s,t \neq i} \frac{\delta(s,t|i)}{\delta(s,t)} \tag{3}$$

According to Table 3, the top five ports in terms of intermediary centrality are Kaohsiung and Hong Kong ports, indicating that these ports are strong intermediaries in the network with strong cargo transshipment capabilities and hub capabilities. However, in terms of the network, there are seventeen ports with zero intermediary centrality, accounting for 20.7%. This finding indicates that there is still some unbalanced development in the RCEP shipping network. In terms of port distribution, the top ports are still distributed in East and Southeast Asia, which is consistent with the results of the two centrality analyses in the previous section, but due to the relatively large number of Shanghai ports as ports of departure, the intermediate centrality rankings have dropped, but they are still in the top 10.

**Table 3.** Betweenness centrality rank.

| Port | Betweenness Centrality | Regional Rank | Network Rank |
|---|---|---|---|
| Kaohsiung | 18.941 | 1 | 2 |
| Hong Kong | 8.623 | 2 | 3 |
| Ningbo | 5.193 | 3 | 6 |
| Shanghai | 5.054 | 4 | 7 |
| Shenzhen | 3.719 | 5 | 10 |

### 3.2.4. Core-Edge Analysis

The core-edge structure is a particular structure consisting of several elements interconnected with each other, with the centre closely connected and the periphery sparsely dispersed. The core degree index reflects the close connection between the network nodes and measures the importance of the nodes in the network, which can reflect the importance of the port nodes in the network from another aspect. This article utilises UCINET software to calculate the ranking of the core degree of the nodes in the network, and the ranking results are shown in Table 4.

**Table 4.** Corene rank.

| Port | Corene | Regional Rank | Network Rank |
|---|---|---|---|
| Hong Kong | 0.267 | 1 | 2 |
| Shenzhen | 0.249 | 2 | 3 |
| Kaohsiung | 0.246 | 3 | 4 |
| Shanghai | 0.237 | 4 | 5 |
| Ningbo | 0.221 | 5 | 6 |
| Qingdao | 0.209 | 6 | 9 |
| Xiamen | 0.205 | 7 | 10 |

As can be seen in Table 4, the centrality of the top ten Chinese ports ranked in the network is all greater than 0.2. In addition, by calculation, the average centrality of the 82 ports is 0.082, and using the average centrality as a boundary, it can be concluded that there are 33 core ports and 49 edge ports, and the centrality of the top-ranked Chinese ports is much higher than the baseline, which is a side effect of proving that the Chinese ports are very important in the network. The results of the core-edge analysis are consistent with the results of the first three centrality analyses, and Chinese ports still occupy the majority of the list, which also shows that Chinese ports are very important to the RCEP shipping network.

*3.3. Port Ranking Based on the Entropy Weight Method*

3.3.1. Method Selection

To make the port ranking in this paper more objective, an assignment method is selected to assign the multi-centrality analysis indicators and finally obtain the comprehensive ranking of port importance. Currently, there are subjective and objective means to assign the indicators, among which are the subjective assignment method, which is the expert survey method, and the objective assignment method, which is the entropy weight method. The entropy weight method is widely used in risk evaluation, a comprehensive evaluation method that can compare and select multiple indicators and effectively compensate for the defects of subjectivity and arbitrariness, particularly in the maritime and port industry [34]

Existing studies regarding the risks of extreme weather on container ports prefer to utilise qualitative research methods, such as the Delphi method and expert survey, to determine the weights of the indicators, which are more subjective, and the indicator weights of the model, which can be affected by subjectivity and objectively respond to the actual risk posed by typhoons. Compared with those qualitative methods, the entropy weight method utilises the characteristics of "entropy" in physics to assign weights to the indicators, which can exclude the influence of some subjective factors and is more conducive to the objective and comprehensive ranking of the ports in the network. The steps and methods of calculating weights are shown in Equations (4)–(7).

Firstly, the values are normalised as

$$x_{ij} = \frac{x_{ij} - \min\{x_{1j}, \cdots, x_{nj}\}}{\max\{x_{1j}, \cdots, x_{nj}\} - \min\{x_{1j}, \cdots, x_{nj}\}} \tag{4}$$

Calculate the weight $P_{ij}$ of the *j*th indicator for the *i*th node as follows:

$$P_{ij} = \frac{x_{ij}}{\sum\limits_{i=1}^{n} x_{ij}} \tag{5}$$

The entropy value $e_j$ of the *j*th indicator is

$$e_j = -\frac{1}{ln(n)} \sum\limits_{i=1}^{n} P_{ij} In(P_{ij}) \tag{6}$$

The weight $w_j$ of the *j*th indicator is

$$w_j = \frac{1 - e_j}{m - \sum e_j} \tag{7}$$

3.3.2. Port Ranking

According to the method of calculating weights in Section 3.3.1, the weights corresponding to the four indicators of degree centrality, proximity centrality, intermediate centrality, and core-edge analysis can be obtained, as shown in Table 5.

**Table 5.** Index weight.

| Degree Centrality | Closeness Centrality | Betweenness Centrality | Corene |
|:---:|:---:|:---:|:---:|
| 22.17% | 3.31% | 55.90% | 18.62% |

According to the weights obtained, the final ranking of each port in the network can be obtained through the port comprehensive ranking formula (Equations (3)–(8)), and the

comprehensive ranking of important ports in the RCEP shipping network is shown in Table 6.

$$Rank = \sum_{j=1}^{4} w_j \cdot R_j \qquad (8)$$

**Table 6.** RCEP shipping network port comprehensive ranking.

| Port | Comprehensive Score | Regional Rank | Network Rank |
|---|---|---|---|
| Kaohsiung | 2.37 | 1 | 2 |
| Hongkong | 2.81 | 2 | 3 |
| Shanghai | 6.08 | 3 | 5 |
| Ningbo | 6.54 | 4 | 7 |
| Shenzhen | 7.23 | 5 | 8 |
| Qingdao | 10.05 | 6 | 9 |

After calculation, the Chinese ports ranked in the top 10 ports in the comprehensive ranking of the RCEP shipping network ports are Kaohsiung Port, Hong Kong Port, Shanghai Port, Ningbo Zhoushan Port, Shenzhen Port, and Qingdao Port. The number that accounts for 60% of the list shows the important position of Chinese ports in the network. Kaohsiung and Hong Kong ports have stable rankings in the multi-centrality analysis and are in the second and third positions in the shipping network rankings. However, the port of Shanghai is ranked third in the regional ranking and fifth in the shipping network, which is different from other statistics where the port of Shanghai has been ranked as the No. 1 global port for twelve consecutive years. The main reason for this is that, in the statistics of shipping routes, the ports that the port of Shanghai connects to are mostly Ningbo Zhoushan and Qingdao ports with a high number of repetitions, which explains that the value of the degree of degree is not as good as that of the other ports in the multi-centrality ranking. The port of Shanghai is also relatively more often ranked as a port in the network than other ports. At the same time, Shanghai port appears more as a port of departure in the network, so its role as a hub port is relatively weakened. At the same time, the ranking only analyses the statistical situation of the connection between the ports and does not include the port throughput in the statistical scope, which leads to the third position of Shanghai port in the regional comprehensive ranking. Ningbo Zhoushan Port, Shenzhen Port, and Qingdao Port are ranked seventh, eighth, and ninth in the shipping network.

## 4. Analysis of the RCEP Shipping Network

The stability of the RCEP shipping network affects the trade between member countries, and when the ports in the network are affected by external factors that lead to failure, it will inevitably have a greater or lesser adverse effect on the network and even affect the stability of the network to a certain extent. Therefore, in this section, the changes in network performance after the failure caused by typhoons of six important ports in China will be evaluated from three aspects: network robustness, average shortest path length, and global network efficiency, and the evaluation results will be analysed accordingly.

### 4.1. Network Robustness Analysis Based on the Relative Size of Maximum Connected Subgraphs

Robustness generally refers to the ability of a system to maintain its performance after a particular sustained impact, which can be used to analyse the stability of the shipping network after the failure of a port. Ports often encounter emergencies or meteorological disasters that lead to the failure of port closure, so the robustness of the network can be tested to find the weak links in the shipping network and take appropriate measures to improve the stability of the network.

In the theory of complex networks, when a node in the network is attacked and fails, the connection between nodes is disconnected, and the original network will be split into several small networks, i.e., connectivity subgraphs, due to the disconnection of connecting

edges. The one that contains the most nodes among these split connectivity subgraphs is called the maximal connectivity subgraph. The relative size of the maximal subgraph, *S*, represents the performance degradation of the network due to external influences and is obtained from the ratio of the number of nodes, $N'$, in the maximal subgraph to the total number of nodes in the network, *N*. This finding is not only an intuitive indicator of the performance degradation of the network due to external influences, it also visualises the number of nodes in the maximum connectivity subgraph and reflects the degree of network damage by external factors. The formula for calculating the relative size of the maximum connectivity subgraph is shown below.

$$S = \frac{N'}{N} \tag{9}$$

To carry out the network robustness test after the six important ports in China are affected, this section carries out two scenarios of deliberate attacks on the six nodes of the important ports in China to examine the changes in the robustness of the network after the nodes fail. The two types of scenarios correspond to the impact of the ports affected by the recovery efficiency of the ports after the impact of the port is fast or slow. One of them is that the ports are affected and fully recovered before the next node is affected, i.e., examining the change in network robustness after the port is affected and resumes operation promptly. The other is the change in network robustness when the port cannot recover after being affected, and the port node fails for a longer period once it fails, i.e., when the port does not recover promptly. Therefore, the experiment needs to make the following assumptions:

1. Node failure is a complete failure, i.e., once the failure disconnects all the connecting edges with other nodes;
2. Node failures are not transmitted;
3. In the just-in-time recovery scenario, nodes are resilient and fully recovered before the next node is affected; in the not-even-if-recovery scenario, nodes are not resilient and fail once they fail;
4. All the failures are caused by typhoons.

According to Figure 2, the network's robustness decreases to 98.72% in the case of timely resumption of port operations, while the robustness of the shipping network further decreases when the ports cannot resume operations on time. When all the essential ports in China are out of service, the robustness of the shipping network decreases to 92.31%, which is far lower than that in the case of timely resumption conditions. If the number of suspended ports rises, the robustness of the shipping network is bound to fall further, thus affecting the stability of the network. Thus, we need to minimise the occurrence of multiple ports failing simultaneously and take specific measures to minimise the affected degree of the ports as much as possible to improve the stability of the shipping network.

*4.2. Average Path Length Analysis*

The average shortest path length represents the average of the shortest path lengths between all nodes in the network. In the network, the shorter the average shortest path is, the better the network connectivity and the higher the efficiency; on the contrary, the worse the connectivity and the lower the efficiency. The average shortest path length of the RCEP shipping network obtained through UCINET software is 2.3, which indicates that the goods in the network need to be transferred at least twice to reach the destination port. When the important ports in China are affected by typhoons, the ports' recovery speed will affect the network's performance, and the average shortest path will change accordingly. Therefore, in this section, assuming single port failure and multiple port failure, corresponding to the cases that the affected ports have recovered before the subsequent port failure and the affected ports have not recovered before the subsequent port failure, respectively, experiments will be conducted to determine the change in the average shortest path of the network, and the results are shown in Figure 3.

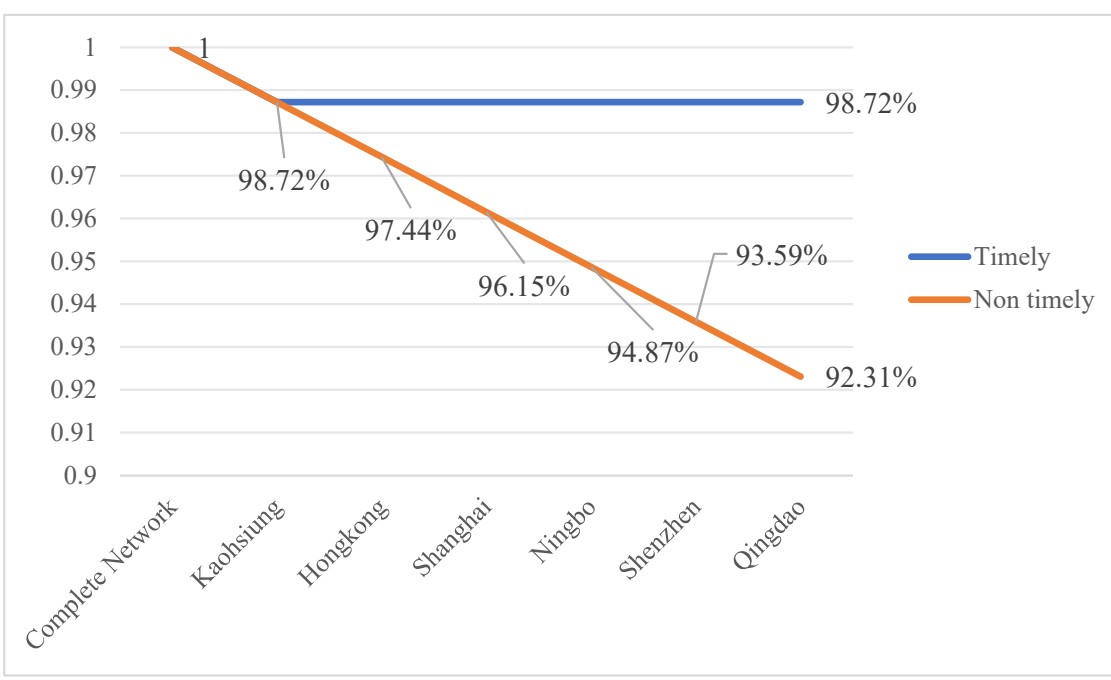

**Figure 2.** Robustness in the two recovery scenarios.

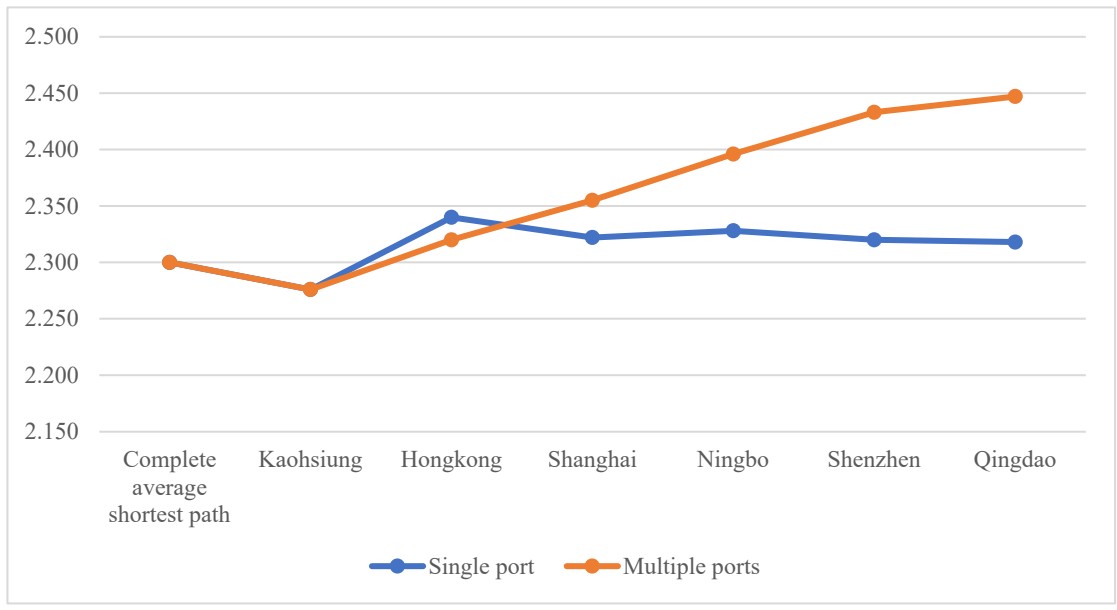

**Figure 3.** Changes in the average shortest path of important ports in China affected by typhoons.

According to the experimental results, it can be seen that when a single port is affected, the average path length of the network fluctuates up and down at 2.3, while when multiple ports are affected, the average path length of the network shows an increase with the number of ports, rising from 2.3 to 2.45.

Comparing the two cases, the average path length of the network under the influence of multiple ports significantly increases compared with that under the influence of a single port. This means that when more than a critical port in the network is affected, the transportation of goods may be transferred more than once to reach the destination port, which leads to a reduction in the transportation efficiency of the network, and the time and cost of the transportation will be significantly increased. Meanwhile, from the perspective of network average path length, it can also be found that the impact of multi-port failure

is more significant than that of single-port failure. Thus, it is necessary to improve the recovery efficiency of ports, change the possible multi-port failure into a single-port failure, minimise the impact due to typhoons, and reduce the fluctuation of the network's average path length.

### 4.3. Global Network Efficiency Analysis

Global network efficiency represents the connectivity between network nodes and the efficiency of the global network, which can better reflect the network's performance. The reciprocal of the distance $d_{ij}$ between any two nodes in the network can be expressed as the connectivity efficiency $f_{ij}$, and the global network efficiency $E_f$ can be obtained by taking the average value of the connectivity efficiency between many nodes. The higher the global network efficiency, the higher the transportation efficiency in the network and the higher the trade efficiency of the network. In this section, the impact of the complete failure of China's important ports in the network on the global network efficiency is analysed by calculating the global network efficiency, which is shown in Equation (10), where $n$ is the total number of nodes in the shipping network. The experimental results of global network efficiency are shown in Figures 4 and 5.

$$E_f = \frac{1}{n(n+1)}\sum_{i \neq j} f_{ij} \tag{10}$$

According to the experimental results shown in Figure 4, when only one important port in the network is affected, the network efficiency shows that the more important the port is, the greater the degree of decline in network efficiency. When the important ports in the network are affected, the network's average efficiency decreases to 47.73%, a decrease in efficiency of about 2.26%. In terms of localisation, when Kaohsiung Port is affected, the network efficiency decreases to the greatest extent, the efficiency decreases to 45.4%, the rest of the ports are affected by the network efficiency decrease to less than 2%, and the impact caused by the closure of a single port is about 5%.

However, when multiple ports in the shipping network are affected, as shown in Figure 5, i.e., multiple port nodes fail at the same time, it can be found that the network efficiency drops from 49.99% to 35.97%, which is a decrease in efficiency of nearly 30%. Compared with a single port, after the impact, the efficiency value is also lower by 11.76%, which has a huge impact on the shipping network; the efficiency of cargo transportation will also be greatly reduced, and the length of transportation is significantly longer. The situation of multiple ports being affected is consistent with the possible impact of typhoon disasters. As typhoons have a large impact area and usually result in several ports in a region being affected simultaneously or sequentially, it is important to shorten the recovery time for ports, as failure to do so would have a huge impact on shipping-related supply chains and international trade and result in huge economic losses.

The absence of a detailed analysis of typhoons and their specific effects on shipping networks suggests a potential area for further research. Given the broad impact of typhoons and their ability to affect multiple ports simultaneously or in quick succession, understanding the specific dynamics of such natural disasters could be crucial for developing more effective preparedness and response strategies. This new study could involve analysing historical data on typhoons' paths, intensity, and impact on shipping networks and modelling future scenarios to inform infrastructure improvements, contingency planning, and policymaking to enhance global shipping networks' resilience to such natural disruptions.

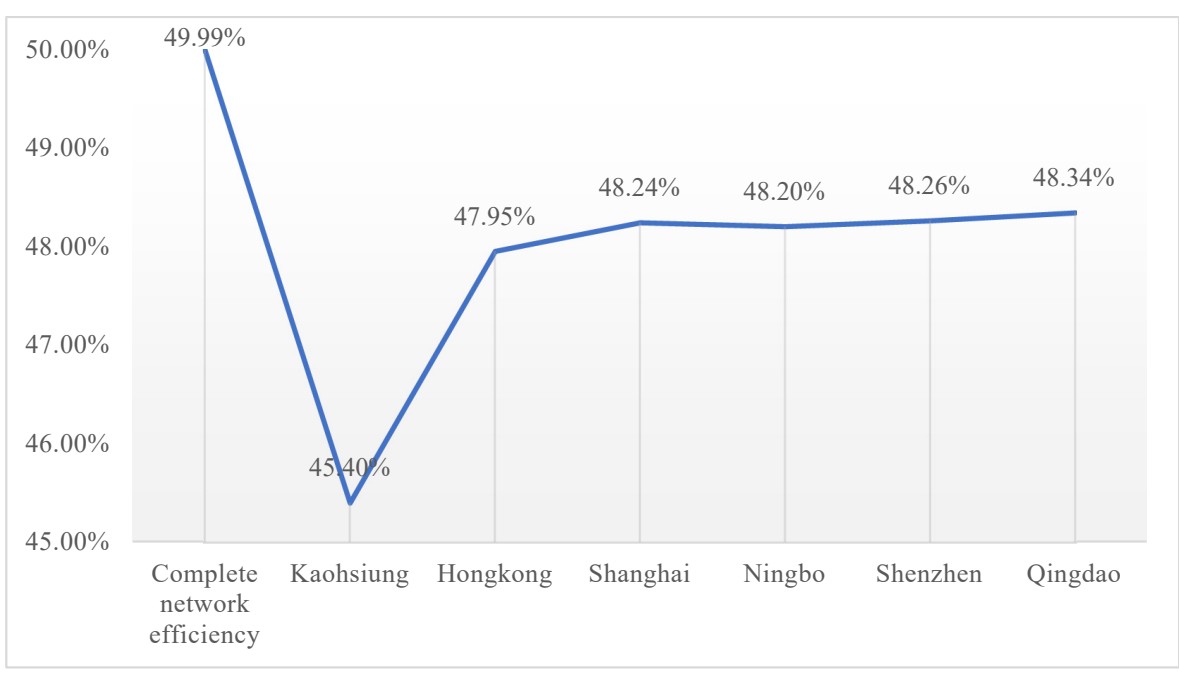

**Figure 4.** Changes in the network efficiency of the RCEP shipping network with a single port affected.

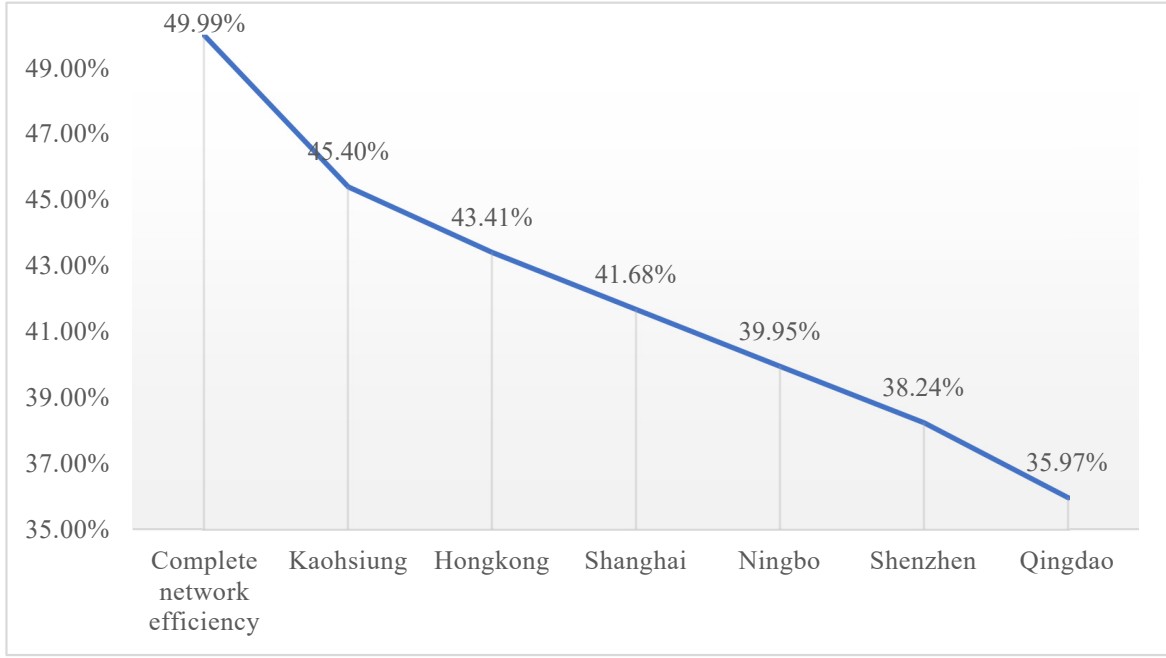

**Figure 5.** Changes in the network efficiency of the RCEP shipping network with multiple ports affected.

## 5. Discussion

Given the results and the newly integrated perspective on sustainability, our study deepens the understanding of network efficiency within the global shipping framework, focusing on the resilience of Chinese ports to the impacts of climate change, including typhoons. By leveraging complex network analysis, this research highlights the pivotal role of Chinese ports in the global shipping network and underscores the need for enhanced resilience and adaptability in response to climatic threats. The integration of sustainability practices, as discussed, extends the application of network theory in maritime logistics and disaster resilience, introducing a nuanced approach to evaluating port efficiency and

robustness amidst environmental and societal challenges. This dual focus enriches the theoretical conversation around supply chain vulnerability and resilience, incorporating sustainability as a crucial factor in the discourse.

From a practical standpoint, the insights from this study, enriched by the emphasis on sustainability practices, offer valuable guidance for port authorities, policymakers, and logistics companies. Recognising the interconnectedness of network efficiency and port disruptions, this research advocates for adopting advanced predictive analytics, infrastructure enhancement, and sustainability integration to counteract the adverse effects of climate disturbances. Furthermore, it highlights the critical importance of regional cooperation, particularly among RCEP member countries, in fostering a collective approach to response and recovery efforts. Such collaborative endeavours are essential for maintaining supply chain stability, minimising economic losses, and advancing the sustainability agenda within the maritime logistics and freight transportation sector.

## 6. Conclusions

In conclusion, our analysis of the RCEP shipping network, focusing on China's crucial ports, underscores the significant vulnerabilities posed by typhoon disasters. The experimental results reveal that the robustness, average shortest path length, and network efficiency of the shipping network are all adversely affected when typhoons impact China's major ports. Notably, performance loss is considerably more severe when multiple ports are affected simultaneously or quickly, a common scenario during typhoon seasons. This multiplicative effect of port failures due to typhoons can lead to substantial operational disruptions, economic losses, and delays in network recovery.

Our research complements and extends existing studies on port efficiency and resilience to natural disasters. While previous works have focused on the impact of specific disasters on port operations and local economies, this study provides a holistic view of network efficiency under the stress of typhoons, particularly within the RCEP region. Unlike the studies that primarily addressed theoretical or localised impacts, our study systematically examines the broader implications of typhoon disruptions on the global shipping network's efficiency, incorporating immediate and long-term effects.

The findings of this study highlight the imperative need to enhance the resilience of the shipping network against typhoon disasters. To achieve this, future research should delve into the development of advanced predictive models and simulations that can accurately forecast typhoons and assess their potential impact on shipping networks. Additionally, there is a need for comprehensive resilience assessment models that encompass both the physical infrastructure and the socio-economic dynamics affecting port recovery times. Comparative analyses with other regions facing similar natural disaster threats could offer valuable insights into effective mitigation and recovery strategies. The mitigation and recovery measures and strategies can be identified as chances to enhance efficiency and generate some profit (i.e., energy efficiency), among other opportunities, like deploying technological innovations, job opportunities, and trade growth [35].

From a policy perspective, governments and port authorities must invest in strengthening and retrofitting port infrastructures to withstand the impacts of typhoons better. This includes physical improvements and integrating advanced technologies, such as the IoT, AI, and blockchain, to enhance sustainability performance monitoring, operational efficiency, better decision making, and recovery post-disaster. Human health risks and life threats can be minimised in the forthcoming years [4]. Moreover, enhancing regional collaboration for disaster preparedness and response among RCEP member countries can play a pivotal role in minimising the impacts of such disasters.

Furthermore, the challenges and solutions identified in this study extend beyond the RCEP region, suggesting a global relevance. Therefore, it is recommended that the principles and strategies developed within the RCEP framework for enhancing shipping network resilience should be considered for global adoption. Extending these practices worldwide could significantly bolster the global shipping industry's ability to withstand

and recover from typhoon impacts, thereby ensuring more stable and efficient supply chains on a global scale. As such, port sustainability dimensions cover not only ports per se but also land transport (i.e., trucks) and vessels. This finding identifies ports' vital role in maritime supply chains and their crucial nodes between land and sea without territorial borders. Additionally, the port's sustainability actions are relevant to all the United Nations (UN) Sustainable Development Goals (SDGs). Ports may highlight their sustainability and connect their actions with all 17 SDGs. The most closed connections are SDG 9, "Sustainable Development Goal", SDG 13, "Climate Action", and SDG 14, "Life Below Water" [36]. In doing so, ports may exhibit their sustainability methods that are different from usual practices. Moreover, using measures, actions, and UN SDGs, ports can develop key performance indicators (KPIs) and sustainability reports to examine sustainability performance. In the meantime, regional port associations and organisations can employ the complete framework to benchmark ports as comparable case studies [35].

In light of the above, reducing the impact of typhoons on ports and shortening recovery times emerge as critical objectives. Achieving these goals will not only mitigate the loss of performance and economic losses caused by typhoons but also strengthen the overall resilience of the shipping network. This effort requires a concerted approach, encompassing advanced research, policy support, and international collaboration, to safeguard against the far-reaching effects of natural disasters on global trade and economic stability.

**Author Contributions:** Conceptualisation, T.W. and M.C.-P.P.; methodology, H.F.; software, H.F.; validation, H.F., M.C.-P.P. and Y.-Y.L.; formal analysis, H.F.; investigation, H.F.; data curation, T.W. and H.F.; writing—original draft, H.F.; writing—review and editing, T.W. and M.C.-P.P.; visualisation, M.C.-P.P.; supervision, T.W.; funding acquisition, T.W. and Y.-Y.L. All authors have read and agreed to the published version of the manuscript.

**Funding:** This work was funded by the Shanghai Pujiang Program (21PJC068) and IEC\NSFC\2231-96—International Exchanges 2022 Cost Share (NSFC), as well as a grant from the Research Grants Council of the Hong Kong Special Administrative Region China (UGC/FDS24/B07/22).

**Institutional Review Board Statement:** Not applicable.

**Informed Consent Statement:** Not applicable.

**Data Availability Statement:** The corresponding author will make the datasets available upon reasonable request.

**Conflicts of Interest:** The authors declare no conflicts of interest.

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
