# Peer review of "Analysis of the Network Efficiency of Chinese Ports in Global Shipping under the Impacts of Typhoons"

_sustainability, doi:10.3390/su16083190_

Round 1

Reviewer 1 Report

Comments and Suggestions for Authors

Using UCINET software, this study constructs a global shipping network and evaluates the centrality of ports by calculating their degree, proximity, and overall centrality scores, subsequently ranking them accordingly. It then identifies the top 20 ports in China based on this ranking and examines their network efficiency while considering the risks posed by typhoons. The analysis encompasses various aspects such as network robustness, average shortest path length, and overall network efficiency, both for the shipping network as a whole and for China's significant port nodes within it.

While the paper is well-written overall, there are several areas for improvement:

1. Although the paper systematically constructs a shipping network based on RCEP and analyzes key ports in China, it is recommended that the research motivation and innovative aspects be explicitly stated at the conclusion of the introduction.

2. In the "Related Work" section, references should be cited using only the authors' surnames, and this section should be subdivided into more detailed subsections to highlight the research motivation effectively.

3. The overall quality of Figure 1 requires improvement.

4. In Section 4, conclusions are drawn from various analyses; however, there is a need for an effectiveness analysis between different models. Additionally, a sensitivity analysis of the parameters used in the analysis tools should be provided.

5. The format of the references does not adhere to journal requirements, and the author should verify essential bibliographic information accordingly.

Author Response

Reviewer 1

  1. Although the paper systematically constructs a shipping network based on RCEP and analyses key ports in China, it is recommended that the research motivation and innovative aspects be explicitly stated at the conclusion of the introduction.

Answer: Thanks for this comment. The research motivation and innovative aspects have been added in the abstract. “It emphasises the significant impact of typhoons on multiple ports and reminds people to minimise losses as much as possible based on experimental results, ensuring the stable operation of ports and improving their resilience in typhoon prevention.”

  1. In the “Related Work” section, references should be cited using only the authors’ surnames, and this section should be subdivided into more detailed subsections to highlight the research motivation effectively.

Answer: Thanks for this comment. We have revised the citation formats and divided the context into two sections: 2.1 Shipping research based on complex networks; 2.2. Shipping research based on typhoon risks

  1. The overall quality of Figure 1 requires improvement.

Answer: Thanks for this comment. Because there are over 80 port nodes and over 800 connected edges, it might not be easy to improve the visibility of this figure again. We have explained in the context. “In this paper, an 82*82 0-1 adjacency matrix was constructed using UNCINET software. Thus, the RCEP shipping network was established based on the composition of a fully connected graph, as shown in Figure 1. The network graph has a total of 82 port nodes and 866 connected edges.”

  1. In Section 4, conclusions are drawn from various analyses; however, there is a need for an effective analysis between different models. A sensitivity analysis of the parameters used in the analysis tools should also be provided.

Answer: The relevant content has been revised in the conclusion.

  1. The format of the references does not adhere to journal requirements, and the author should verify essential bibliographic information accordingly.

Answer: Thank you. We have changed the format of the references to adhere to journal requirements.

Reviewer 2 Report

Comments and Suggestions for Authors

I appreciate the opportunity to review the manuscript titled “Analysis of Network Efficiency of Chinese Ports in Global Shipping Under the Impacts of Typhoon”. I extend my congratulations to the authors for their diligent research. While the article brings forth valuable insights and important considerations, I believe there are areas that require improvement. Below are some recommendations to enhance the manuscript.

In the abstract, it is essential for authors to distinctly articulate the objective of the article, the methodologies employed, and the outcomes achieved.

There is a need to clarify the titles of sections 2.1. and 2.2, they have the same names?

A discussion section is absent. I recommend incorporating a discussion part that emphasizes both theoretical and practical implications, elucidating how this study contributes to the existing literature, and providing a comparison with research conducted by other scholars.

The reference list should adhere to the formatting guidelines specified by the journal.

Upon reviewing the manuscript, it appears that it may not align with the criteria for publication in Sustainability journal as it lacks emphasis on sustainability aspects and fails to establish a clear connection to them.

Author Response

Reviewer 2

Below are some recommendations to enhance the manuscript.

  1. In the abstract, it is essential for authors to distinctly articulate the objective of the article, the methodologies employed, and the outcomes achieved.

Answer: Thank you very much. The abstract is revised.

  1. There is a need to clarify the titles of sections 2.1. and 2.2, they have the same names?

Answer: Thanks for this comment. We have revised the two sections: 2.1 Shipping research based on complex networks; 2.2. Shipping research based on typhoon risks

  1. A discussion section is absent. I recommend incorporating a discussion part that emphasises both theoretical and practical implications, elucidating how this study contributes to the existing literature and providing a comparison with research conducted by other scholars.

Answer: Thank you. We have added a discussion section to indicate theoretical and practical implications.

  1. The reference list should adhere to the formatting guidelines specified by the journal.

Answer: We have changed the format of the references to adhere to journal requirements.

  1. Upon reviewing the manuscript, it appears that it may not align with the criteria for publication in Sustainability journal as it lacks emphasis on sustainability aspects and fails to establish a clear connection to them.

Answer: Thank you. We have revised the paper thoroughly and indicated the connection between this work and sustainability in the abstract, introduction, discussion and conclusion.

Reviewer 3 Report

Comments and Suggestions for Authors

First of all, I would like to say that the authors have chosen an interesting topic and delivered an interesting paper.

However, there are some important parts missing that I would like to comment on.

First of all, I do not really see the reasons why this article should be published in Sustainability. It is clearly devoted to transport infrastructure problems and would be better suited for specific journals, such as Logistics. There is not even one single mention of sustainability problems in the paper.

If the authors want to be published in a Sustainability journal, they need to add more material about the influence of transport infrastructure on sustainable development.

Regarding the contents of the paper.

The authors declare that they aim to analyze the network efficiency of Chinese Ports under the impacts of typhoons. Yet, there is no analysis of typhoons and their effects. The authors talk about typhoons in the Introduction and Literature Review, totally ignore them in the Results, and talk again in the Conclusion.  

Author Response

Reviewer 3

First of all, I would like to say that the authors have chosen an interesting topic and delivered an interesting paper. However, there are some important parts missing that I would like to comment on.

  1. First of all, I do not really see the reasons why this article should be published in Sustainability. It is clearly devoted to transport infrastructure problems and would be better suited for specific journals, such as Logistics. There is not even one single mention of sustainability problems in the paper. If the authors want to be published in a Sustainability journal, they need to add more material about the influence of transport infrastructure on sustainable development.

Answer: Thank you. We have revised the paper thoroughly and indicated the connection between this work and sustainability in the abstract, introduction, discussion and the conclusion.

  1. Regarding the contents of the paper. The authors declare that they aim to analyse the network efficiency of Chinese Ports under the impacts of typhoons. Yet, there is no analysis of typhoons and their effects. The authors talk about typhoons in the Introduction and Literature Review, totally ignore them in the Results, and talk again in the conclusion. 

Answer: Thank you. We have analysed the impact of a single port disruption and the impact of multiple port disruptions at the end of Section 4. Also, I have written a paragraph regarding the potential future developments.

Reviewer 4 Report

Comments and Suggestions for Authors

The authors have crafted a thought-provoking piece that delves into the intricacies of freight demand modeling. However, several areas require attention to elevate the paper to its full potential.

Firstly, there is a need for internationalization as the current focus on China-centric sources limits the breadth of perspectives. Sources in the literature need to be diversified – almost all of them have Chinese authors! - and discussing results beyond the Chinese context would enrich the paper's relevance and applicability on a global scale.

Secondly, apart from its geography, the literature review would benefit from refinement, particularly by incorporating a comprehensive discussion on freight demand uncertainty. Integrating uncertainty within the model framework will enhance the paper's robustness and address a critical aspect of freight transportation analysis. Suggested sources: Aguas, O., Bachmann, C. (2022). Assessing the effects of input uncertainties on the outputs of a freight demand model.

Additionally, Figure 1 requires better depiction to ensure clarity and readability for readers. A revised visualization would facilitate understanding and interpretation of the presented data.

Furthermore, the authors should provide rationale for choosing entropy as the method for weight selection, acknowledging alternative approaches and explaining the suitability of entropy in their context.

Lastly, a brief discussion on the implications of the study's findings on sustainability would align the paper more closely with the journal's thematic focus. The name of this journal is “Sustainability”, so that sustainability considerations are paramount in the research appearing on it, and their inclusion would enhance the paper's overall contribution to the field. Sources: Fulzele, V., Shankar, R. (2022). Improving freight transportation performance through sustainability best practices.

In conclusion, while the paper shows promise, it requires significant revisions to fulfill its potential. With these adjustments, the paper could merit a second chance and make a meaningful contribution to the discourse on freight transportation and sustainability.

Comments on the Quality of English Language

No major issues detected

Author Response

Reviewer 4

The authors have crafted a thought-provoking piece that delves into the intricacies of freight demand modeling. However, several areas require attention to elevate the paper to its full potential.

  1. Firstly, there is a need for internationalisation as the current focus on China-centric sources limits the breadth of perspectives. Sources in the literature need to be diversified – almost all of them have Chinese authors! - and discussing results beyond the Chinese context would enrich the paper’s relevance and applicability on a global scale.

Answer: Thank you. We have revised the Literature review thoroughly.

  1. Secondly, apart from its geography, the literature review would benefit from refinement, particularly by incorporating a comprehensive discussion on freight demand uncertainty. Integrating uncertainty within the model framework will enhance the paper’s robustness and address a critical aspect of freight transportation analysis. Suggested sources: Aguas, O., Bachmann, C. (2022). Assessing the effects of input uncertainties on the outputs of a freight demand model.

Answer: Thank you. We have added it to the literature review as well.

  1. Additionally, Figure 1 requires better depiction to ensure clarity and readability for readers. A revised visualisation would facilitate understanding and interpretation of the presented data.

Answer: Thanks for this comment. Because there are over 80 port nodes and over 800 connected edges, it might not easy to improve the visibility of this figure again. We have explained in the context. “In this paper, an 82*82 0-1 adjacency matrix was constructed using UNCINET software. Thus, the RCEP shipping network was established based on the composition of a fully connected graph, as shown in Figure 1. The network graph has a total of 82 port nodes and 866 connected edges.”

  1. Furthermore, the authors should provide rationale for choosing entropy as the method for weight selection, acknowledging alternative approaches and explaining the suitability of entropy in their context.

Answer: Thanks for this comment. We have further explained reason of choosing entropy method in 3.3.1.

“The entropy weight method is widely used in risk evaluation, which is a comprehensive evaluation method that can compare and select multiple indicators, and also effectively compensate for the defects of subjectivity and arbitrariness, in particular in maritime and port industry (e.g., Ren, 2020).

Existing studies regarding the risks of extreme weather on container ports prefer to utilise qualitative research methods, such as the Delphi method and expert survey to determine the weights of the indicators, which is more subjective, and the indicator weights of the model, which can be affected by subjectivity and objectively respond to the real risk posed by typhoons. Compared with those qualitative methods, the entropy weight method utilises the characteristics of “entropy” in physics to assign weights to the indicators, which can exclude the influence of some subjective factors, which is more conducive to the objective and comprehensive ranking of the ports in the network”

  1. Lastly, a brief discussion on the implications of the study’s findings on sustainability would align the paper more closely with the journal’s thematic focus. The name of this journal is “Sustainability”, so that sustainability considerations are paramount in the research appearing on it, and their inclusion would enhance the paper’s overall contribution to the field. Sources: Fulzele, V., Shankar, R. (2022). Improving freight transportation performance through sustainability best practices.

Answer: Thank you. We have added it to the literature review as well. Also, I have added a discussion section for academic and practical implications for improving freight transportation performance through sustainability best practices.

  1. In conclusion, while the paper shows promise, it requires significant revisions to fulfill its potential. With these adjustments, the paper could merit a second chance and make a meaningful contribution to the discourse on freight transportation and sustainability.

Answer: Thank you. We have revised the paper thoroughly and indicated the connection between this work and sustainability in the abstract, introduction, discussion and the conclusion.

Round 2

Reviewer 3 Report

Comments and Suggestions for Authors

The authors have successfully covered the main issues that I have highlighted in my previous review.

I don't have any more questions.